# Balancing Token Efficiency and Structural Accuracy in LLMs Image Generation by Combining VQ-VAE and Diffusion Tokenizers

## Abstract

We proposes a novel visual tokenizer by combining high-level semantic tokens and low-level pixel tokens to represent images, aiming to address the challenges of image-to-sequence conversion for Large Language Models (LLMs). Existing visual tokenizers, such as VQ-VAE and diffusion-based models, either struggle with token explosion as image resolution increases or fail to capture detailed structural information. Our method introduces a dual-token system: high-level semantic tokens capture the main content of the image, while low-level pixel tokens preserve structural details. By integrating these tokens in a hybrid architecture, we leverage a VQ-VAE branch to generate low-resolution guidance and a diffusion process to reconstruct high-resolution images with both semantic coherence and structural accuracy. This approach significantly reduces the number of required tokens and enhances image reconstruction quality, offering an efficient solution for tasks like image generation and understanding based on LLMs.

## 1 Introduction

In natural language processing, Large Language Models (LLMs) have demonstrated exceptional performance, largely due to the Scale law [Kaplan et al. (2020); Henighan et al. (2020); McCandlish et al. (2018)], which allows them to effectively learn relationships within long sequences. This has enabled them to excel in various text-based tasks. If other modalities can also be converted into sequences, LLMs could become a powerful tool for multimodal tasks. For example, if images are transformed into a sequence of tokens, LLMs could learn the relationships between these tokens, enabling tasks like image generation and understanding.

A visual tokenizer converts images into token sequences, aiming to achieve this conversion with minimal cost. The goal is to use as few tokens as possible while retaining the essential information during the transformation. The tokenizer typically consists of an encoder, decoder, and a codebook. The encoder transforms the image into a sequence of embeddings, which are then converted into discrete tokens using the codebook. The decoder reconstructs the image from the token sequence. The similarity between the reconstructed image and the original is a direct measure of how much information is lost during the conversion. Significant differences indicate a larger information loss.

For LLMs, maintaining information fidelity during the image-to-sequence conversion is crucial. Excessive information loss can negatively impact downstream tasks, such as poor alignment between generated images and text in image generation tasks, or a lack of detail comprehension in image understanding tasks.

Currently, two main types of visual tokenizers exist: VQ-VAE-based [Van Den Oord et al. (2017)] and diffusion-based [Nichol & Dhariwal (2021)] approaches. VQ-VAE methods divide an image into patches, which are processed by CNNs or ViTs to convert each patch into a token. While this method retains the overall structure of the image, the number of tokens increases exponentially with image resolution, making LLM learning more difficult. Moreover, the fixed number of tokens limits the generated image to a fixed resolution, reducing flexibility. In contrast, diffusion-based tokenizers use tokens that encode high-level semantic information, disregarding pixel-level details. The number of tokens remains low, and the image resolution is independent of the token count, but the reconstructed image may differ from the original in structural details, such as posture.

| Method | #Token | Token sematic | Flexibility | Similarity |
|--------|--------|---------------|-------------|------------|
| VQ-VAE | more | Pixel Level | No | Yes |
| Diffusion | less | High Level | Yes | No |
| Ours | less | High + Pixel Level | Yes | Yes |

Table 1: Comparison between VQ-VAE based Tokenizer and Diffusion based Tokenizer

Inspired by these methods, we propose a dual-token approach: one set of tokens represents high-level semantic information, while another set captures lower-level pixel details to control the image structure. For instance, in Figure 4, the high-level semantic tokens could describe the main content, such as "A Doberman Pinscher lounging on a sunny porch with its tongue out, next to a brick wall and a white door." However, these tokens might omit finer details like the dog's fur patterns, specific posture, or the exact position of the wall and door. These details can be roughly captured by low-level pixel tokens.

To achieve this dual-token representation and reconstruct the image, we designed the architecture shown in Figure 1. We use a SEED [Ge et al. (2023)] encoder to obtain high-level semantic tokens and a VQ-VAE (MoVQ [Zheng et al. (2022)]) branch to extract low-level pixel tokens. The final image is generated through a diffusion process: a low-resolution image is created using pixel tokens, noise is added, and the denoising process begins. High-level semantic tokens guide this process to reconstruct the image.

This design offers several advantages: 1) VQ-VAE only needs to generate a low-resolution guide image, so the number of tokens required (around 40-300) is far fewer than with traditional VQ-VAE methods (which often need thousands of tokens). This advantage becomes more pronounced when generating ultra-high-resolution images, as diffusion reconstruction is resolution-independent. 2) Our method solves the problem of structural differences between the original and reconstructed images when using only high-level semantic tokens. It achieves high similarity to the original image with minimal additional token cost.

## 2 RELATED WORKS

In recent years, the field of image and video generation has seen substantial progress, primarily driven by two major approaches: diffusion models and visual tokenization combined with large language models (LLMs).

### 2.1 DIFFUSION MODELS

Diffusion models have emerged as a leading method for high-quality image and video synthesis. The foundational work by Ho et al. (2020) on Denoising Diffusion Probabilistic Models (DDPM) introduced a process where images are generated through iterative denoising from a noisy starting point, setting a new standard in generative models. Building on this, subsequent research has focused on improving the efficiency and quality of these models. Notably, Improved DDPM [Nichol & Dhariwal (2021)] and ADM [Song et al. (2020)] refined the denoising process and loss functions, leading to more stable and higher-fidelity outputs.

Further advancements include the introduction of Latent Diffusion Models (LDM) by Rombach et al. (2022), which significantly enhanced computational efficiency by applying the diffusion process in a latent space rather than the pixel space. This approach has been successfully implemented in widely used models such as Stable Diffusion. More recently, GenTron [Chen et al. (2024)] has integrated Transformer architectures with diffusion models, achieving state-of-the-art results in text-to-image and text-to-video generation tasks, demonstrating the potential of diffusion models in handling complex generative tasks.

### 2.2 VISUAL TOKENIZATION AND LLMS

Parallel to the development of diffusion models, visual tokenization combined with large language models has emerged as another promising approach. This method typically involves encoding visual

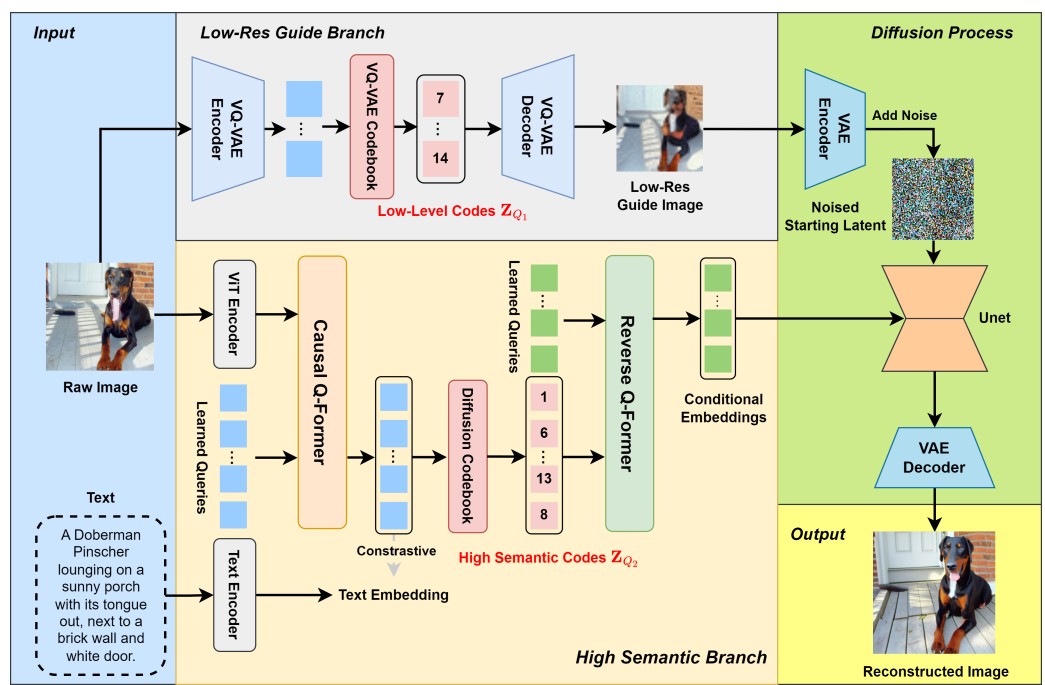

Figure 1: Overview of the Ours architecture.

information into discrete tokens, which are then processed by language models to generate new images or videos. The VQ-VAE [Van Den Oord et al. (2017)] framework laid the groundwork for this approach by introducing a method to quantize image representations into discrete latent spaces.

Building on VQ-VAE, DALL-E [Ramesh et al. (2021)] and VQGAN [Esser et al. (2021)] demonstrated the potential of combining these tokenized representations with transformer-based models to generate high-quality images from text prompts. More recent efforts, such as OmniTokenizer, have pushed this concept further by creating a unified tokenization framework that can handle both images and videos. OmniTokenizer [Wang et al. (2024)] utilizes a spatial-temporal Transformer architecture, enabling the simultaneous processing and tokenization of image and video data, which can then be used in both language models and diffusion models for generation tasks.

### 2.3 VISUAL TOKENIZATION APPROACHES

Visual tokenization has followed two primary paths: those based on VQ-VAE and those involving diffusion-based reconstruction of quantized images. The VQ-VAE-2 [Razavi et al. (2019)] model advanced the original VQ-VAE by introducing multi-scale architectures, improving the resolution and detail in generated images. OmniTokenizer further extended this by introducing a progressive training strategy that allows for effective spatial and temporal representation learning, applicable to both images and videos. Recently, TiTok model [Yu et al. (2024)] introduced a more efficient 1D tokenization approach, which significantly enhanced both image reconstruction and generation tasks. On the other hand, diffusion-based tokenization approaches, exemplified by SEED [Ge et al. (2023)] and LaVIT [Jin et al. (2023)], reconstruct images using diffusion processes, only need fewer tokens than VQ-VAE based methods, achieving higher efficiency and quality.

### 3 METHOD

Our proposed method is structured three primary components: **Low-Res Guide Branch**, **High Semantic Branch** and **Diffusion Branch**. This section will discuss the design and function of each branch in detail.

### 3.1 Branch I: Low-Res Guided Branch

In this branch, we have emulated the design of MoVQ [Zheng et al. (2022)], witch is a state-of-the-art VQ-VAE based tokenizer, with modifications tailored to enhance performance for our specific task. The structure of this branch consists of three primary modules: the Encoder, the Vector Quantizer, and the Decoder.

#### 3.1.1 Encoder

The Encoder is responsible for compressing the input image $\mathbf{X} \in \mathbb{R}^{H \times W \times C}$, where $H$ and $W$ represent the height and width of the image, and $C$ represents the number of color channels (typically 3 for RGB images). We increased the compression rate of the Encoder compared to the MoVQ to extract more relevant information from high-resolution images.

$$\mathbf{Z}_E = \text{Encoder}(\mathbf{X}) \in \mathbb{R}^{H' \times W' \times D} \tag{1}$$

Here, $\mathbf{Z}_E$ is the latent representation, $H'$ and $W'$ are the downscaled dimensions, and $D$ is the dimensionality of the latent space.

#### 3.1.2 Vector Quantizer

The latent representation $\mathbf{Z}_E$ is then passed through a Vector Quantizer, which discretizes $\mathbf{Z}_E$ into a set of discrete latent codes $\mathbf{Z}_{Q_1}$. The quantization process maps each vector in $\mathbf{Z}_E$ to the closest vector in a codebook of learnable embeddings.

$$\mathbf{Z}_{Q_1} = \text{Quantize}(\mathbf{Z}_E) \in \mathbb{R}^{H' \times W' \times D} \tag{2}$$

$\mathbf{Z}_{Q_1}$ is the quantized latent representation, which is the token sequence used for image generation.

#### 3.1.3 Decoder

The Decoder is responsible for reconstructing the low-res guide image from the quantized latent codes $\mathbf{Z}_{Q_1}$. Unlike MoVQ, we adopted a smaller upsampling rate in the Decoder. This modification is intended to reconstruct the original image $\hat{\mathbf{X}}$ that has been resized to a smaller dimension.

$$\hat{\mathbf{X}} = \text{Decoder}(\mathbf{Z}_{Q_1}) \in \mathbb{R}^{\hat{H} \times \hat{W} \times C} \tag{3}$$

Here, $\hat{H}$ and $\hat{W}$ represent the dimensions of the reconstructed image, which are smaller than the original dimensions $H$ and $W$.

#### 3.1.4 Fine-Tuning

We fine-tuned this branch on ImageNet for several epochs. During this process, we accounted for the difference in size between the reconstructed image and the original image. Since the Decoder reconstructs an image of size $\hat{H} \times \hat{W}$, which is smaller than the original image size $H \times W$, the original image $\mathbf{X}$ needs to be resized to match the dimensions of the reconstructed image before calculating the loss.

Specifically, if the original input image size is $256 \times 256$, and the reconstructed image size is $64 \times 64$, the original image $\mathbf{X}$ is first resized to the $64 \times 64$ resolution, resulting in a resized image $\mathbf{X}_{\text{resized}}$.

$$\mathbf{X}_{\text{resized}} = \text{Resize}(\mathbf{X}, (\hat{H}, \hat{W})) \in \mathbb{R}^{\hat{H} \times \hat{W} \times C} \tag{4}$$

We then compute the reconstruction loss between the resized original image $\mathbf{X}_{\text{resized}}$ and the reconstructed image $\hat{\mathbf{X}}$. The reconstruction loss is typically measured using mean squared error (MSE):

$$\mathcal{L}_{\text{recon}} = \frac{1}{\hat{H} \times \hat{W} \times C} \sum_{i=1}^{\hat{H} \times \hat{W} \times C} (\mathbf{X}_{\text{resized},i} - \hat{\mathbf{X}}_i)^2 \tag{5}$$

This approach ensures that the reconstructed image at a lower resolution retains as much fidelity as possible to the resized original image, thereby enhancing the accuracy and quality of the reconstruction.

## 3.2 BRANCH II: HIGH SEMANTIC BRANCH

The High Semantic Branch is designed following SEED [Ge et al. (2023)] to represent an image as a sequence of discrete visual codes with high-level semantics. The tokenizer is also based on a vector quantization (VQ) framework and employs an encoder-decoder architecture to achieve this representation. The branch comprises a ViT image encoder, a Causal Q-Former, a VQ codebook and a Reverse Q-Former.

Initially, a pre-trained ViT image encoder, derived from the BLIP-2 model, processes the input image to generate a 2D feature representation arranged as 16×16 tokens. The Causal Q-Former then converts these features into a sequence of 32 tokens with causal dependencies, representing the high-level semantic content of the image in a 1D format. Subsequently, the VQ codebook quantizes these causal embeddings into 32 discrete visual codes. These codes maintain the causal structure and are well-suited for capturing the semantic essence of the image. Finally, the Reverse Q-Former decodes these visual codes into 77 generation embeddings, which is used as the condition for diffusion process.

### 3.2.1 CAUSAL Q-FORMER

The training process for the Causal Q-Former is illustrated in Figure 1. A set of 32 learnable query embeddings, along with the image features produced by the pre-trained ViT encoder, are fed into the Causal Q-Former. The Causal Q-Former employs self-attention layers with a causal mask, allowing each query embedding to interact only with previous queries, and cross-attention layers, enabling interaction with the image features. The training is conducted using contrastive learning, fine-tuning the Causal Q-Former on 5 million image-text pairs. The objective is to maximize the similarity between the final causal embedding and the corresponding text features while minimizing the similarity with text features from other image-text pairs in the batch. The contrastive loss function is given by:

$$\mathcal{L}_{\text{contrastive}} = -\frac{1}{N} \sum_{i=1}^{N} \left[ \log \frac{\exp(\text{sim}(\mathbf{z}_i, \mathbf{t}_i)/\tau)}{\sum_{j=1}^{N} \exp(\text{sim}(\mathbf{z}_i, \mathbf{t}_j)/\tau)} \right] \tag{6}$$

where $\mathbf{z}_i$ represents the $i$-th causal embedding, $\mathbf{t}_i$ is the corresponding text feature, and $\tau$ is a temperature parameter.

### 3.2.2 QUANTIZATION AND DE-TOKENIZATION

In the second stage, the causal embeddings are quantized into discrete visual codes $\mathbf{Z}_{Q_2}$ using the VQ codebook. This quantization process involves mapping each causal embedding to its nearest neighbor in the codebook. The quantized codes are then decoded back into continuous causal embeddings using a multi-layer Transformer decoder. During training, the cosine similarity between the decoder output and the original causal embeddings is maximized:

$$\mathcal{L}_{\text{cosine}} = -\frac{1}{N} \sum_{i=1}^{N} \frac{\mathbf{z}_i \cdot \hat{\mathbf{z}}_i}{\|\mathbf{z}_i\| \|\hat{\mathbf{z}}_i\|} \tag{7}$$

where $\hat{\mathbf{z}}_i$ represents the reconstructed embedding output by the decoder.

Subsequently, the Reverse Q-Former decodes the discrete visual codes into generation embeddings, which are further optimized to align with the text features of a frozen Stable Diffusion model using mean squared error (MSE) loss:

$$\mathcal{L}_{\text{MSE}} = \frac{1}{N} \sum_{i=1}^{N} \|\mathbf{g}_i - \mathbf{t}_i\|^2 \tag{8}$$

where $\mathbf{g}_i$ denotes the generation embedding, and $\mathbf{t}_i$ is the corresponding text feature.

Through these two stages of training, the Diffusion tokenizer branch effectively aligns image and text features, maintaining semantic consistency, which is crucial for both generation and other downstream tasks.

### 3.3 Branch III: Diffusion Branch

In this stage, we start with VQ-VAE Guided image $\hat{\mathbf{X}}$, which is first encoded into a latent representation $\mathbf{z}_0$ using a Variational Autoencoder (VAE). Next, we apply a Diffusion model with DDIM inversion to add noise to this latent space. This noisy latent representation $\mathbf{z}_T$ becomes the starting point for the denoising process.

The DDIM inversion process introduces noise to the latent space step-by-step, represented as:

$$\mathbf{z}_t = \sqrt{\alpha_t}\mathbf{z}_{t-1} + \sqrt{1-\alpha_t}\epsilon_\theta(\mathbf{z}_{t-1}, t) \tag{9}$$

where $\mathbf{z}_{t-1}$ is the latent variable at step $t-1$, $\alpha_t$ is the noise schedule parameter at time step $t$, and $\epsilon_\theta(\mathbf{z}_{t-1}, t)$ is the predicted noise at this step. The noise is gradually added to transform $\mathbf{z}_0$ into $\mathbf{z}_T$.

Once the noisy latent $\mathbf{z}_T$ is obtained, it serves as the initial state for the Diffusion process. Under the guidance of conditional semantic tokens $\mathbf{g}$, the latent is gradually denoised step-by-step to recover the original structure. The denoising process at each step $t$ can be expressed as:

$$\mathbf{z}_{t-1} = f(\mathbf{z}_t, \mathbf{g}, t) + \epsilon \tag{10}$$

where $f(\mathbf{z}_t, \mathbf{c}, t)$ is the denoising function guided by the condition $\mathbf{g}$, and $\epsilon$ is the residual noise. Through this iterative process, we recover the final latent variable $\mathbf{z}_0'$, which is decoded back into a high-resolution image $\hat{x}$ using the VAE decoder.

This process ensures that the generated image adheres to the guiding conditions while preserving the structural integrity of the original image.

## 4 Experiment

### 4.1 Comparison with SOTA Methods

We have presented a comparison of our model with several state-of-the-art (SOTA) models based on diffusion, with qualitative comparisons illustrated in Figure 2. It can be observed that SEED and Lavit, though preserving semantic similarity with the original image during reconstruction, show significant differences in structure (such as pose, orientation, color, etc.). After structural guidance, our model shows notable improvement in structural reconstruction compared to the former two. Table 2 displays the comparisons across several pixel-level reconstruction metrics, including SSIM, and PSNR. It's important to note that SSIM and PSNR are more inclined to assess pixel-level reconstruction similarity. Since most diffusion-based models often forego pixel-level reconstruction in favor of high-level semantic consistency, these two metrics are not particularly meaningful for those models. However, since VQVAE-based models aim for pixel-level reconstruction, comparing these two metrics is relevant. Nonetheless, due to our model's increased structural similarity with the original image in its reconstruction results, it has also achieved better performance than SEED and Lavit, and close to VQ-VAE based tokenizers.

Table 3 compares the semantic alignment between images and text restored by different tokenizers. We used 5,000 image-text pairs from the COCO dataset, with captions serving as prompts, and evaluated the restored images. The metrics compared were PickScore [Kirstain et al. (2023)], ImageReward [Xu et al. (2023)], and HPSV2 [Wu et al. (2023)]. The results show that our method achieves better semantic alignment with the text.

Since pixel-level restoration comparisons in Table 2 are not very meaningful for diffusion based tokenizers, we conducted a user voting experiment to compare the restoration results of our model with those of SEED and LAVIT, both diffusion-based tokenizers. We randomly sampled 25 sets of images from ImageNet, and users ranked them based on similarity to the original images, assigning scores of 3, 2, and 1 to the first, second, and third places, respectively. Fifteen volunteers participated in the voting, and the results are shown in Table 4. It is evident that our method achieves significantly higher image similarity compared to the other two methods. The 25 sets of images used in this voting can be found in the appendix.

### 4.2 Ablation of Different Resolution of Guide Image

The higher the resolution of the guiding images produced by VQ-VAE, the closer the final images generated by the Diffusion process will be to the original in terms of detail. However, the clearer

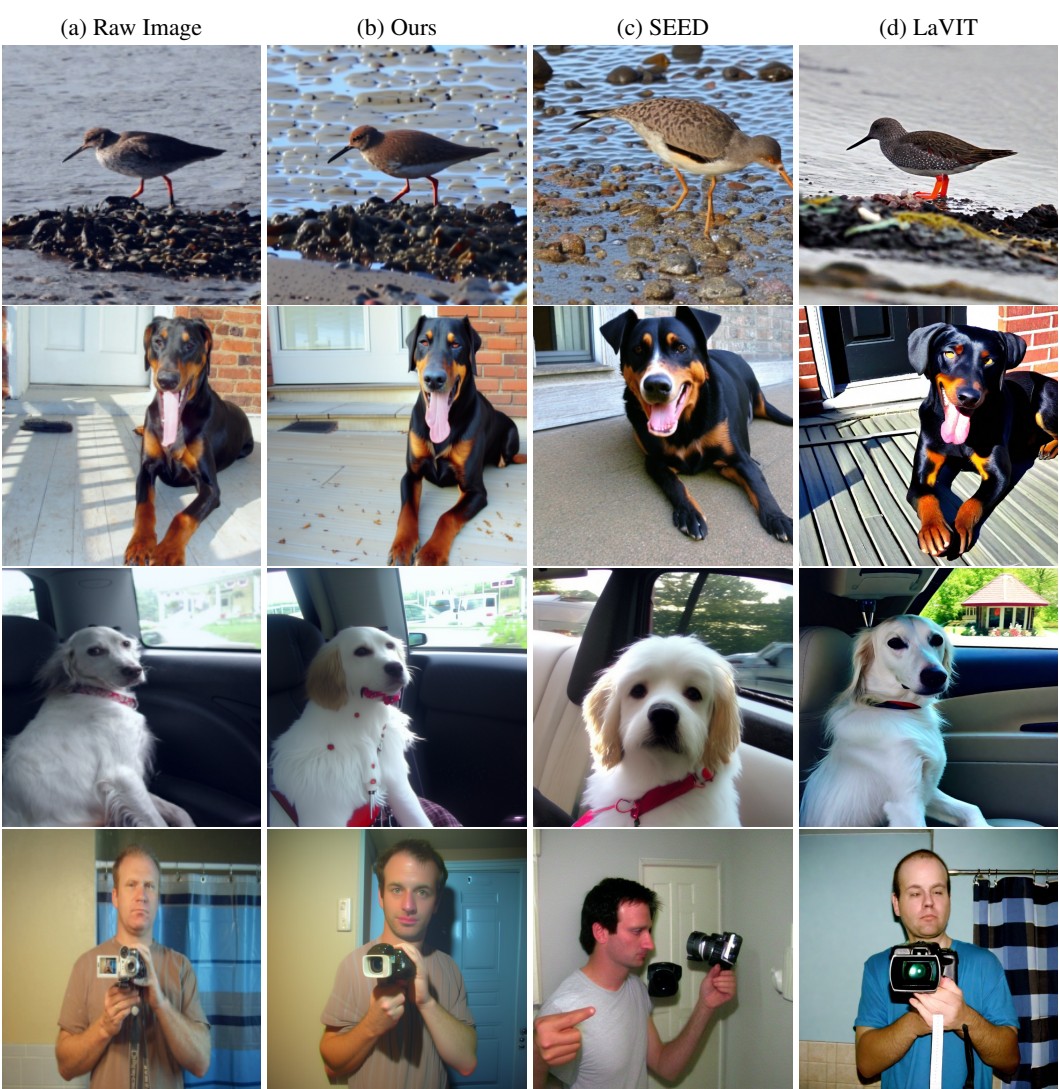

|  | (a) Raw Image | (b) Ours | (c) SEED | (d) LaVIT |

Figure 2: Qualitative Comparisons with SEED and LaVIT

| Method | SSIM | PSNR | comprassing rate | #Tokens |
|---|---|---|---|---|
| MoVQ Zheng et al. (2022) | 0.52 | 27.57 | 8*8 | 2700 |
| VAR-VAE Tian et al. (2024) | 0.48 | 24.77 | 16*16 | 675 |
| VQGAN Esser et al. (2021) | 0.58 | 28.29 | 8*8 | 2700 |
| MAGVIT2 Yu et al. (2023) | 0.38 | 23.94 | 16*16 | 675 |
| SEED Ge et al. (2023) | 0.002 | 9.12 | fix #tokens | 32 |
| LaVIT Jin et al. (2023) | 0.005 | 8.87 | range | 260-784 |
| Ours | 0.33 | 23.23 | fix #tokens | 372 |

Table 2: Comparison with SOTA Methods

the guiding images are, the more tokens they require, which increases the learning difficulty for the LLM. This section compares the impact of using guiding images with different resolutions on the reconstruction process. Guiding images with resolutions of 128, 64, 32, and 16 require 256, 64, 16, and 4 tokens, respectively. The intuitive comparison of the reconstruction results is shown in Figure 3. It can be observed that, even under the guidance of extremely low-resolution images, the reconstructed images still retain the main structure and bear a resemblance to the original images.

| Method | ImageReward | PickScore | HPSV2 |
|---|---|---|---|
| SEED Ge et al. (2023) | 27.32 | 0.35 | 18.65 |
| LAVIT Jin et al. (2023) | 38.54 | 0.20 | 21.47 |
| Ours | 43.28 | 0.45 | 24.32 |

Table 3: Text Image Align Comparison

| Method | SEED | LAVIT | Ours |
|---|---|---|---|
| Avg Score | 1.22 | 1.90 | 2.88 |

Table 4: User Study

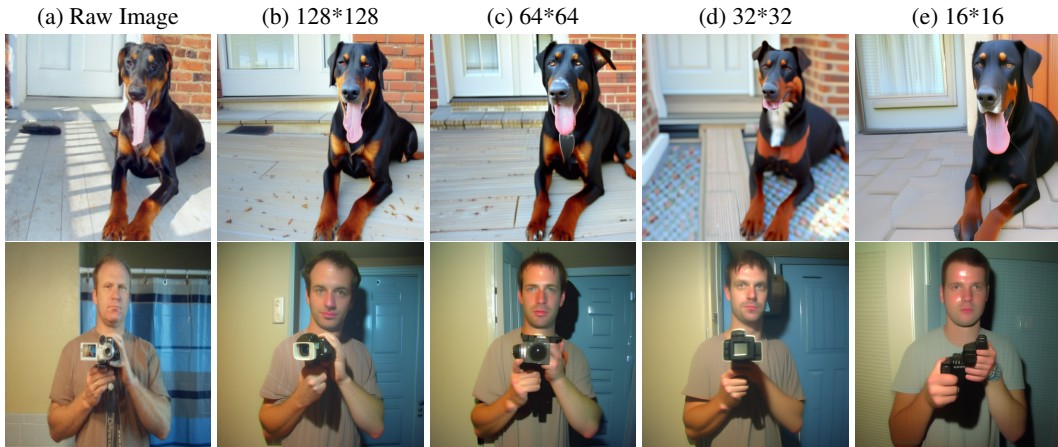

Figure 3: Ablation on Using Different Resolution Guide Images for Reconstruction.

This indicates that only a minimal cost is necessary to correct the structural changes during the reconstruction process by the Diffusion.

### 4.3 COMPARISON OF OUR VQ-VAE GUIDE BRANCH WITH MOVQ

In this section, we compare the reconstruction performance of the VQ-VAE branch and MoVQ in generating structure-guided images. Although Diffusion resampling can tolerate a certain degree of distortion in the guided images, higher quality guided images result in resampled images that are closer to the original. Figure 4 presents a comparison of guided images generated at different resolutions. For ease of display, all guided images were resized to 512x512. It is evident that when reconstructing relatively high-resolution guided images, our model achieves clearer detail reconstruction (e.g., facial features such as eyes in Figure 4). At extremely low resolutions, MoVQ almost entirely loses the structural information of the original image, while our model still retains the general structure of the original image.

### 4.4 COMPARISON WITH SUPER RESOLUTION METHODS

In this section, we compare the method Real-ESRGAN [Wang et al. (2021)] using Super Resolution (SR) to upscaling guide image with our method using Diffusion to re-rendering. Figure 5 shows the results of different resolution guide images, outputted by our VQ-VAE, after being processed by either SR method or Diffusion reconstruction. It is evident that when the resolution of the guide image is relatively high, the SR method performs better. However, as the resolution of the guide image decreases, the performance of the SR method significantly deteriorates, while Diffusion re-rendering still manages to restore the basic structure of the objects. Additionally, Diffusion not only sharpens the guide images but also has a powerful ability to correct structural errors. To demonstrate this, we

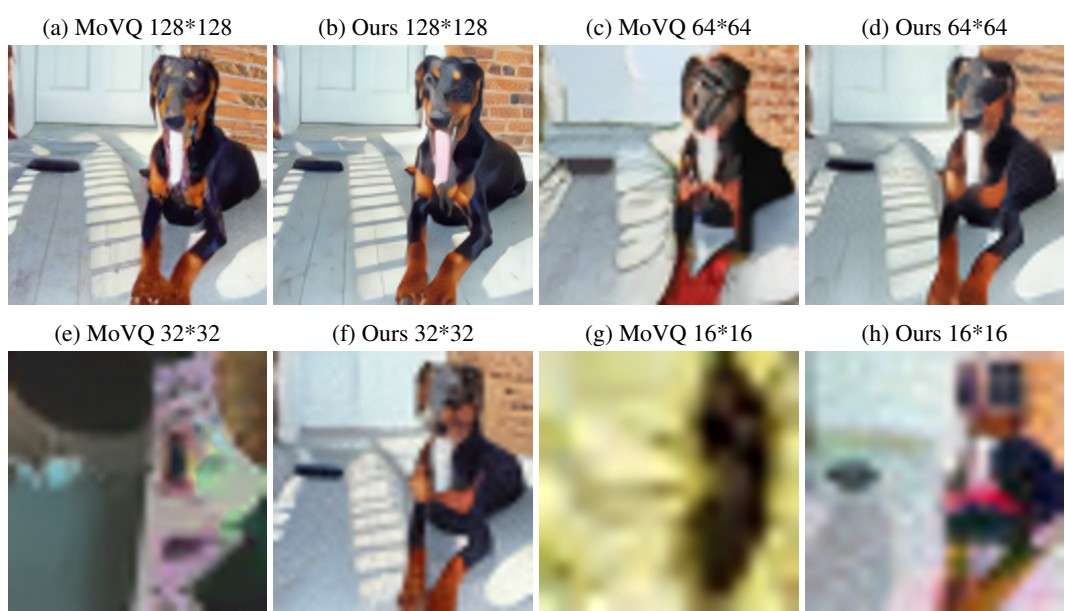

(a) MoVQ 128*128    (b) Ours 128*128    (c) MoVQ 64*64    (d) Ours 64*64

(e) MoVQ 32*32    (f) Ours 32*32    (g) MoVQ 16*16    (h) Ours 16*16

Figure 4: Different Resolution Guide Images

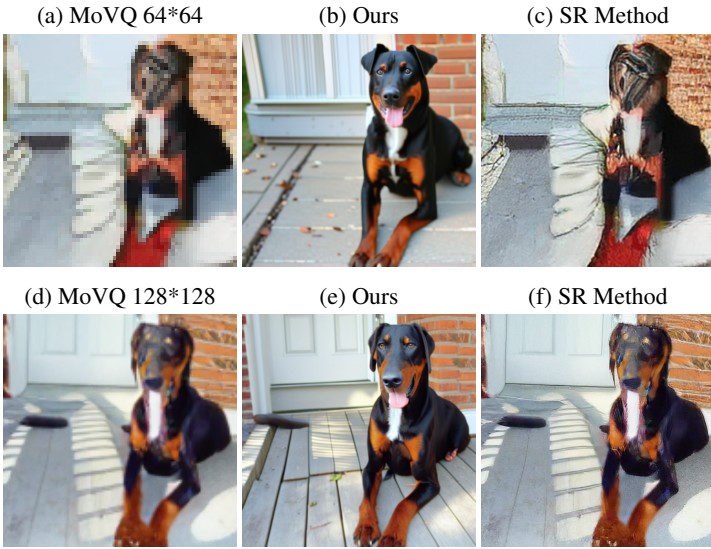

(a) MoVQ 64*64    (b) Ours    (c) SR Method

(d) MoVQ 128*128    (e) Ours    (f) SR Method

Figure 5: Comparison with Super Resolution Methods.

used images with structural errors outputted by MoVQ (as shown in the figure 5, the dog's face is completely unrecognizable). It can be observed that the Diffusion method still reconstructs the correct structure, whereas the SR method makes the structural errors more pronounced. This capability of Diffusion is crucial because LLMs can easily make mistakes in the number or order of tokens when generating outputs, leading to structural anomalies in the images reconstructed by VQ-VAE. Diffusion re-rendering based on high-level semantics can greatly mitigate this issue. Moreover, the number of tokens generated by our model for the guide images is significantly fewer compared to images generated by the VQVAE Tokenizer alone, which greatly reduces the learning difficulty for LLMs and, in turn, reduces the likelihood of structural distortions.

## 5 CONCLUSION

In this study, we reached the following conclusions: (1) The tokens used to represent images can be decomposed into two key components: high-level semantic information and pixel detail information. This decomposition allows for a significant reduction in the number of tokens while maintaining the stability of the restored structure. (2) The Diffusion model can use high-level semantic tokens as conditional input and low-level pixel structure information as the initial latent, thereby generating the restored image.

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
