# OpenReview forum: "Balancing Token Efficiency and Structural Accuracy in LLMs Image Generation by Combining VQ-VAE and Diffusion Tokenizers"
_ICLR.cc/2025/Conference — Submitted to ICLR 2025_

### Official Review · Reviewer_2pmb · 2024-11-02

**Soundness:** 2
**Presentation:** 2
**Contribution:** 2
**Rating:** 3
**Confidence:** 4

**Summary:**

This paper introduces a visual tokenizer by creating a dual-token mechanism that represents images with both high-level semantic tokens and low-level pixel tokens. The method pipeline includes three branches: Low-res guide branch, high semantic branch, and diffusion branch. Evaluation shows that the proposed method has better reconstruction performance compared with SEED and LaViT.

**Strengths:**

- By introducing a low-res guide branch, it improves the reconstruction ability compared with SEED and LaViT.

**Weaknesses:**

My primary concern is that, while this paper introduces a visual tokenizer, it does not demonstrate its usage and effectiveness in image generation. A tokenizer’s utility is limited if it cannot effectively support image generation, and it is unclear how the proposed tokenizer might be used for this purpose. Given the presence of both low-level pixel tokens and high-level semantic tokens, it raises the question of whether both sets of tokens need to be generated simultaneously, and if so, how this generation process would be coordinated.

Apart from the main concern:
- In the introduction, the classification of visual tokenizers is inaccurate. Both “VQ-VAE-based” and “diffusion-based” tokenizers you mention utilize discrete image features, yet many tokenizers employ continuous features. Additionally, the term “diffusion-based” is ambiguous and not representative.
- Regarding the High Semantic Branch, compared to SEED, there is an additional Reverse Q-Former. What is the purpose of this component? I did not find any ablation studies addressing this.
- Furthermore, the writing contains numerous typos, which detracts from the overall quality.

Overall, I think this paper is below the acceptance bar.

**Questions:**

See weakness.

---

### Official Review · Reviewer_yqum · 2024-11-04

**Soundness:** 1
**Presentation:** 1
**Contribution:** 1
**Rating:** 1
**Confidence:** 4

**Summary:**

This paper proposes a hybrid approach to image tokenization that combines VQ-VAE and SEED tokenizers for diffusion models, aiming to balance token efficiency with structural accuracy. The method uses a three-branch architecture: a low-resolution VQ-VAE guide branch, a high-level semantic branch, and a diffusion branch for final image reconstruction, requiring 372 tokens per image. The authors evaluate their approach using both quantitative metrics (SSIM, PSNR) and a user study, comparing against tokenization methods.

Overall, I think this paper was submitted in a rush, which is evident even in the title (which is not very comprehensible) and the first sentence (typo: "we proposes") of the abstract. I strongly suggest that this paper be significantly rewritten and go through another round of review.

**Strengths:**

- This paper identifies a challenge in visual tokenization.
- The authors address this challenge by combining multiple frameworks.

**Weaknesses:**

1. Fundamental Misunderstandings
- The paper misclassifies diffusion models as tokenizers: "two main types of visual tokenizers exist: VQ-VAE-based and diffusion-based approaches"
- The authors appear to confuse tokenization, reconstruction, and generation. For example, SEED and LaVIT are frameworks for understanding and generating images; using and evaluating them for a reconstruction task is fundamentally unfair.

2. Overcomplicated Architecture
- This work employs three branches for sample reconstruction:
  * "Low-Res Guide Branch"
  * "High Semantic Branch"
  * "Diffusion Branch"
- The justification for these branches is unconvincing, especially since the final generation uses Stable Diffusion.
- Several claims are incorrect or overstated:
  * The claim that "the number of tokens increases exponentially with image resolution" is incorrect (it increases quadratically)
  * The statement that "diffusion-based tokenizers use tokens that encode high-level semantic information, disregarding pixel-level details" is generally false
- The paper lacks novel architectural contributions, as each branch is simply derived from previous work.

3. Poor Results

- From Table 2:
| Method | SSIM  | PSNR  |
|-|-|-|
| MoVQ   | 0.52  | 27.57 |
| VQGAN  | 0.58  | 28.29 |
| Ours   | 0.33  | 23.23 |

- Their method significantly underperforms compared to simpler approaches.
- Qualitatively, the reconstructed samples in Fig. 2 show artifacts and significant deviations from the original samples.

4. Misleading Comparisons
- The authors compare their work with SEED/LaVIT while acknowledging different metrics: "SSIM and PSNR are not particularly meaningful for those models"
- They nevertheless use these comparisons to claim improvement. Additionally, comparing with models not trained for reconstruction tasks is methodologically unsound.

5. Questionable Claims

- Their claim: "Our method solves the problem of structural differences between the original and reconstructed images"
- Reality: Their results (0.33 SSIM) demonstrate worse structural preservation than simpler methods like VQGAN (0.58 SSIM)

**Questions:**

1. Architecture:
- Why claim "This design offers several advantages" when the results show worse performance?
- How does adding VQ-VAE actually help when the final generation is still done by diffusion?

2. Results:

Quote: "our model achieves clearer detail reconstruction"
- Where is the evidence for this given the SSIM/PSNR scores?

3. Methodology:
- Why do they say their method needs "minimal cost" when they're using three complex branches, each derived from previous work?
- How do they justify comparing with SEED/LaVIT when they acknowledge the metrics and the task aren't appropriate?

---

### Official Review · Reviewer_QC3E · 2024-11-04

**Soundness:** 2
**Presentation:** 2
**Contribution:** 2
**Rating:** 3
**Confidence:** 4

**Summary:**

The paper proposes combining VQ-VAE and SEED (semantic - diffusion) style image token representations. It proposes a method to represent an image by a combination of VQ-VAE tokens at a low resolution and SEED tokens. An image is reconstructed by first reconstructing its low-resolutoin version from VQ-VAE, and using that as an initialization for diffusion based reconstruction using SEED.

The paper shows that the reconstructed images are better quality than with VQ-VAE alone, and are more 'aligned' with the original image than with SEED alone.

**Strengths:**

The discussion about the relative benefits of SEED and VQ-VAE is insightful.

**Weaknesses:**

The paper combines two existing token representation frameworks. Nevertheless, this could still have been a novel contribution, except that the paper fails to make the connection of how this would be beneficial for LLM based image generation _or_ analysis.

Note that in previous works that the paper compares to --- Lavit and SEED --- token representations are considered as part of a larger system to generate images from text. Reconstruction of the original image is used as an intermediate test, but reconstructing the original images is not the goal of these representations. (Note, these representations are not useful for true image file compression --- there is an entirely separate body of work on neural approaches for image compression.)

The paper does not at all discuss, let alone ablate, how this two-phase token representation would be used in an actual image generation setting. And as far as I can tell, any advantage shown in reconstruction over SEED (which is the second half of the paper's proposed pipeline) --- that it does not 'align well' with the original image --- would be moot:

1. When generating images from pure text, there is no original image.
2. When generating images from text + image input, why not use the image input itself, instead of first compressing and reconstructing it using phase 1 VQ-VAE.

Therefore, the paper in its current form does not make a case for why this token representation is actually useful for Image + LLM applications. It is possible that it has some benefit, but the paper needs a significant rewrite to articulate that benefit, and experiments to confirm it.

**Questions:**

How do you envision this token representation being used in actual text to image generation settings?

---

### Official Review · Reviewer_28yH · 2024-11-04

**Soundness:** 2
**Presentation:** 2
**Contribution:** 1
**Rating:** 3
**Confidence:** 4

**Summary:**

- The paper presents a tokenizer with the goal of reducing the number of tokens required per image. They achieve this by first downsampling the input image (creating a low-resolution copy) which would lead to fewer tokens when mapped by a VAE Encoder. These low-resolution image tokens are appended with semantic-based tokens (semantic-based because they align well with text embeddings). A diffusion-based image reconstruction or denoising objective is used to learn the tokenizer by implicitly performing image super-resolution of the low-resolution image.
- The low-resolution image is generated by performing VQGAN encoding of the image and then decoding back the quantized embedding.
- The semantic embeddings are generated by a SEED (prior work) inspired encoder. Conditioned on the semantic embedding, the low-resolution image is denoised using diffusion UNet architecture.

**Strengths:**

- The paper writing is simple and easy to follow overall.
- The paper attempts to study an interesting topic of mapping images to much fewer tokens (not bounded to patches by using Q-former) with the inspiration of being used for downstream vision-language tasks.

**Weaknesses:**

- Firstly, the use of the word “LLM” in the title and in the method section title (and other places) seems irrelevant as the paper is not at all about LLMs. The only language part is the text-embedding used to train the vision tokenizer. Please remove all occurrences of “LLM” other than motivating the need for fewer tokens for vision-language tasks.
- The overall pipeline is too complicated and probably not a scalable approach for learning compressed embeddings:
    - Firstly, for the desired outcome of generating a low-resolution image, why run a VQGAN encoder/ decoder pipeline and not some simple interpolation-based down-sampler or a small learned down-sampler.
    - For learning the semantic embedding, building on top of prior work SEED, the proposed approach first learns quantized 1D embeddings using Causal Q-Former, but then also performs the task of reverse Q-former, potentially to map quantized embeddings to continuous embeddings. Why not simply learn continuous embeddings from Q-former and not perform any quantization at all? What's the use of quantization and then de-quantization? Also, what's the benefit of such an approach for learning semantic embedding compared to CLIP-based text-image alignment which seems much simpler?

- Lack of novelty or interestingness - On a high level,  the paper mainly augments the SEED tokenizer with an additional low-resolution copy of the input image. The choices made for different component of this pipeline rather made it much more complicated to be useful in practice.
- The approach of reducing the token count per image by downsampling the input and then using a text-aligned semantic embedding as a guide to cover for the resolution and detail loss does not seem to be the right way of learning compressed embeddings. For example –  such an approach (downsampling the image) will lead to a significant loss of details for a highly detailed image, making it difficult to run downstream tasks like VQA on the reconstructed images. Ideally, representation compression for images should be variable like the long-history of work on variable-rate compression.
- Only 4 different images are used to showcase the results of the proposed approach. Visual examples on lot more images would be appreciated to really review the success of the trained model.

**Questions:**

Overall, I would appreciate the author's insights on the particular design choices made for different components of the pipeline (addressing my concern of complicated design choices).

Furthermore, more visual examples will help to better judge the efficacy of the proposed approach.

---

### Meta-Review · Area_Chair_WECz · 2024-12-21

**Metareview:**

This paper proposes a hybrid approach to image tokenization that combines VQ-VAE and SEED tokenizers for diffusion models, aiming to balance token efficiency with structural accuracy. The proposed method uses a three-branch architecture: a low-resolution VQ-VAE guide branch, a high-level semantic branch, and a diffusion branch for final image reconstruction. While the attempt to study a challenging topic is appreciated, the reviews are unanimously negative.  The reviewers raised concerns regarding novelty and validation. Too complicated pipeline is not justified and questionable for scalability.  Significance of introduced two-phase token representation is not convincing.  Effectiveness of introduced visual tokenizer is not demonstrated in image generation. Presented comparison with other methods is not fair.  Writing issues are also pointed out.  Unfortunately, the authors did not submit their rebuttal to address these concerns.  This paper should be rejected, accordingly.

**Additional Comments On Reviewer Discussion:**

See above.

---

### Decision · Program_Chairs · 2025-01-22

Reject